# Effects of the Kampo medicine Yokukansan for perioperative anxiety and postoperative pain in women undergoing breast surgery: A randomized, controlled trial

Moegi Tanaka[1], Tsunehiko Tanaka[2], Misako Takamatsu[3], Chieko Shibue[3], Yuriko Imao[4], Takako Ando[3,4], Hiroshi Baba[1], Yoshinori Kamiya[1]*

1 Division of Anesthesiology, Niigata University Graduate School of Medical and Dental Sciences, Niigata, Japan, 2 Educational Psychology Course, Faculty of Education, Niigata University, Niigata, Japan, 3 Department of Anesthesiology, Niigata Cancer Center Hospital, Niigata, Japan, 4 Department of Anesthesiology, Niigata University Medical and Dental Hospital, Niigata, Japan

* y-kamiya@med.niigata-u.ac.jp

## Abstract

Yokukansan (YKS) is a traditional Japanese herbal (Kampo) medicine prescribed for anxiety. In this randomized controlled trial, we compared the subjective assessment of anxiety using questionnaires and its objective assessment using salivary alpha-amylase concentrations in YKS and control (CNT) groups of women undergoing breast surgery. The trial was registered at the University Hospital Medical Information Network Clinical Trials Registry (registration number: UMIN000028998), and the investigators were blinded to drug administration. One hundred patients who underwent breast cancer surgery were allocated to either the YKS or the CNT group. Finally, 35 and 42 patients in the YKS and CNT groups were analyzed, respectively. The YKS group received two 2.5 g doses of the medication before sleeping on the night before surgery and 2 h before inducing anesthesia, while the CNT group did not receive medication preoperatively. Patients answered two questionnaires, the Hospital Anxiety and Depression Scale and the State-Trait Anxiety Inventory, pre-and postoperatively as subjective anxiety assessments. As an objective anxiety indicator, salivary alpha-amylase levels were measured the day before, directly before, and the day after surgery (T3). In the YKS group, salivary alpha-amylase scores directly before operation were significantly lower than those on the day before surgery and at one day postoperatively (F [2,150] = 3.76, p = 0.03). Moreover, the Hospital Anxiety and Depression Scale-Anxiety and State-Trait Anxiety Inventory-Trait scores were significantly more improved postoperatively in the YKS group than in the CNT group (difference in Hospital Anxiety and Depression Scale-Anxiety: YKS, mean -2.77, 95% confidence interval [-1.48 –-4.06], p <0.001, and CNT, -1.43 [-0.25–-2.61], p = 0.011; and difference in State-Trait Anxiety Inventory: YKS group, -4.23 [-6.95–-1.51], p = 0.0004; and CNT group, 0.12 [-2.36–2.60], p = 0.92). No side effects were associated with YKS. YKS may reduce perioperative anxiety in patients undergoing surface surgery.

**Data Availability Statement:** All relevant data are within the paper and its Supporting Information files.

**Funding:** This work was supported by the Japan Society for the Promotion of Science, Tokyo, Japan (URL: https://kaken.nii.ac.jp/en/) [grant numbers 19K18288 to MT] The funders had no role in study design, data collection and analysis, decision to publish, or preparation of the manuscript.

**Competing interests:** The authors have declared that no competing interests exist.

## Introduction

Perioperative anxiety is associated with unfavorable physiological responses, such as tachycardia and hypertension [1]; it has been shown to prolong postoperative pain [2] and increase the risk of postoperative delirium [3]. Previous studies have shown that preventing preoperative anxiety improves surgical outcomes, including postoperative pain management [4] and recovery from anesthesia [5]. Benzodiazepines effectively reduce perioperative anxiety; however, they are associated with undesirable sedative effects [6]. Furthermore, antidepressants and other anxiolytics are unsuitable for perioperative administration due to their ineffectiveness and side effects, including nausea, thirst, and drowsiness [7].

Postoperative pain is among the most unpleasant symptoms associated with surgery. Severe postoperative pain may increase chronic postsurgical pain (CPSP) incidence. Accordingly, perioperative pain management is considerably important for both patient satisfaction and CPSP prevention. Since postoperative pain is exacerbated by perioperative anxiety, improving anxiety may improve CPSP.

Yokukansan (YKS) is a traditional Japanese herbal (Kampo) medicine that contains seven herbal extracts: Atractylodes Lancea Rhizome, Poria Sclerotium, Cnidium Rhizome, Uncaria Hook, Japanese Angelica Root, Bupleurum Root, and Glycyrrhiza. It is prescribed as a safe and effective medication for treating anxiety symptoms, such as irritability, restlessness, and insomnia. YKS has also been reported to effectively suppress chronic pain in several experimental settings [8] and improve pain disorders, such as headaches [9] and neuropathic pain [10]. Although the mechanism of action of YKS remains unclear, its anxiolytic-like effects may be due to its pharmacological effects on serotonin (5-hydroxytryptamine [5-HT]) and glutamate-mediated nervous system functions [11]. Furthermore, YKS has shown significantly better outcomes than diazepam in the modified Observer's Assessment of Alertness/Sedation Scale in patients who underwent colon surgery [12]. However, no effects were observed on preoperative anxiety and postoperative delirium in patients undergoing highly invasive major surgeries in a study assessing the subacute effects of YKS on preoperative anxiety, that is, the effects of three doses daily for 7 days [13]. Moreover, postoperative delirium may be associated with the duration of surgery, postoperative rest, hypothermia, and intraoperative blood loss [14], which may have influenced the aforementioned study results. The acute effects of two doses of YKS preoperatively and that of YKS on perioperative anxiety and postoperative pain in patients undergoing surface surgery remain unclear.

We designed a randomized controlled trial (RCT) to compare the subjective assessment of anxiety using the Hospital Anxiety and Depression Scale (HADS) and State-Trait Anxiety Inventory (STAI) questionnaires with an objective assessment using measured salivary alpha-amylase (sAA) concentrations in YKS-treated and control groups of women undergoing breast surgery. The visual analog scale (VAS) for pain and quality of recovery (QoR)-15 scores was also measured to compare subjective pain and perioperative recovery quality assessments between both groups.

## Materials and methods

### Study setting and patients

This was a single-blind RCT in which the investigators were blinded to drug administrations. This study was approved by the institutional review board of the Niigata University Medical and Dental Hospital (reference number: 1031190049), and written informed consent was obtained from all enrolled patients. The trial was registered at the University Hospital Medical Information Network Clinical Trials Registry (registration number: UMIN000028998;

principal Investigator: Hiroshi Baba; date of registration: July 10, 2017 [https://upload.umin.ac.jp/cgi-open-bin/ctr/ctr_view.cgi?recptno=R000032363]), and Japan Registry of Clinical Trials (registration number: CRB3180025; principal Investigator: Hiroshi Baba; date of registration: July 2, 2019 [https://jrct.niph.go.jp/latest-detail/jRCT1031190049]) with some modifications: the objective parameter of perioperative anxiety (sAA) was set as the primary outcome and questionnaire-assessed anxiety and perioperative pain (HADS, STAI, QoR-15, and VAS) were set as secondary outcomes. The trial was registered before initiation and patient enrollment. This manuscript adheres to the applicable CONSORT guidelines.

The study included 91 women aged 20–60 years who had been diagnosed with breast cancer and underwent partial or total mastectomy under general anesthesia at the Niigata University and Niigata Cancer Center Hospital between July 2017 and May 2020. Exclusion criteria were an American Society of Anesthesiology-Physical Status (ASA-PS) >4; a body mass index (BMI) >30 kg/m$^2$; preexisting pain anywhere in the body; prescribed analgesics, opioids, antipsychotic drugs, or Kampo medicine; allergy to any drugs or food; psychological problems; central nervous system dysfunction; inability to understand Japanese; and the need for axillary lymph node clearance by intraoperative rapid pathological examination due to positive sentinel lymph node metastasis.

Participants were randomly assigned to the treatment (YKS) or control (CNT) groups using a computerized dynamic allocation algorithm written by the Division of Psych Statistics of Niigata University. Allocation was adjusted by minimization for procedure and age.

## Intervention

One day preoperatively (T1), the age, BMI, surgical history, and ASA-PS of all eligible patients were recorded. On the night before surgery, HADS, STAI, and QoR-15 questionnaire surveys were conducted, and sAA was measured.

Patients in the YKS group received two doses of 2.5 g Tsumura Yokukansan Extract Granules (TJ-54, Tsumura & Co., Tokyo, Japan) before sleeping on the night before surgery and 2 h before inducing anesthesia, and 2.5 g of YKS contains 1.083 g of a dry extract of mixed herbal medicines in the following proportions: Atractylodes Lancea Rhizome 19.5%, Poria Sclerotium 19.5%, Cnidium Rhizome 14.6%, Uncaria Hook 14.6%, Japanese Angelica Root 14.6%, Bupleurum Root 9.7%, and Glycyrrhiza 7.3%. Patients in the CNT group only received water. sAA was measured again immediately preoperatively and immediately upon arrival to the operating room (T2). All eligible patients were induced with propofol-based general anesthesia, and only remifentanil was used as an intraoperative narcotic. One day postoperatively (T3), sAA was measured; VAS was assessed; HADS, STAI, and QoR-15 questionnaires were conducted again. The questionnaire was administered by an anesthesiologist who was not involved in the division of study groups and was not informed to which group the patients were assigned. The study outline is shown in Fig 1.

We measured sAA as an objective indicator of anxiety since psychosocial stress increases its release, reflecting the activity of the sympathetic-adrenal-medullary (SAM) system [15]; consequently, this system is associated with anxiety and arousal [16]. sAA was measured using a hand-held monitor (COCORO meter, NIPRO, Osaka, Japan) consisting of a disposable test strip and a monitor. The test strip contains 2-chloro-4-nitrophenyl-4-o-b-D-galactopyranosyl-maltoside [17], which is hydrolyzed by alpha-amylase [18]. The hydrolyzing reaction proceeds until the substrates are completely consumed. The monitor quantifies the amylase activity by assessing the reaction time. sAA quantity has been suggested to reflect stress-related physiological changes and is used as an index for psychological stress, including anxiety, fear, and frustration [19].

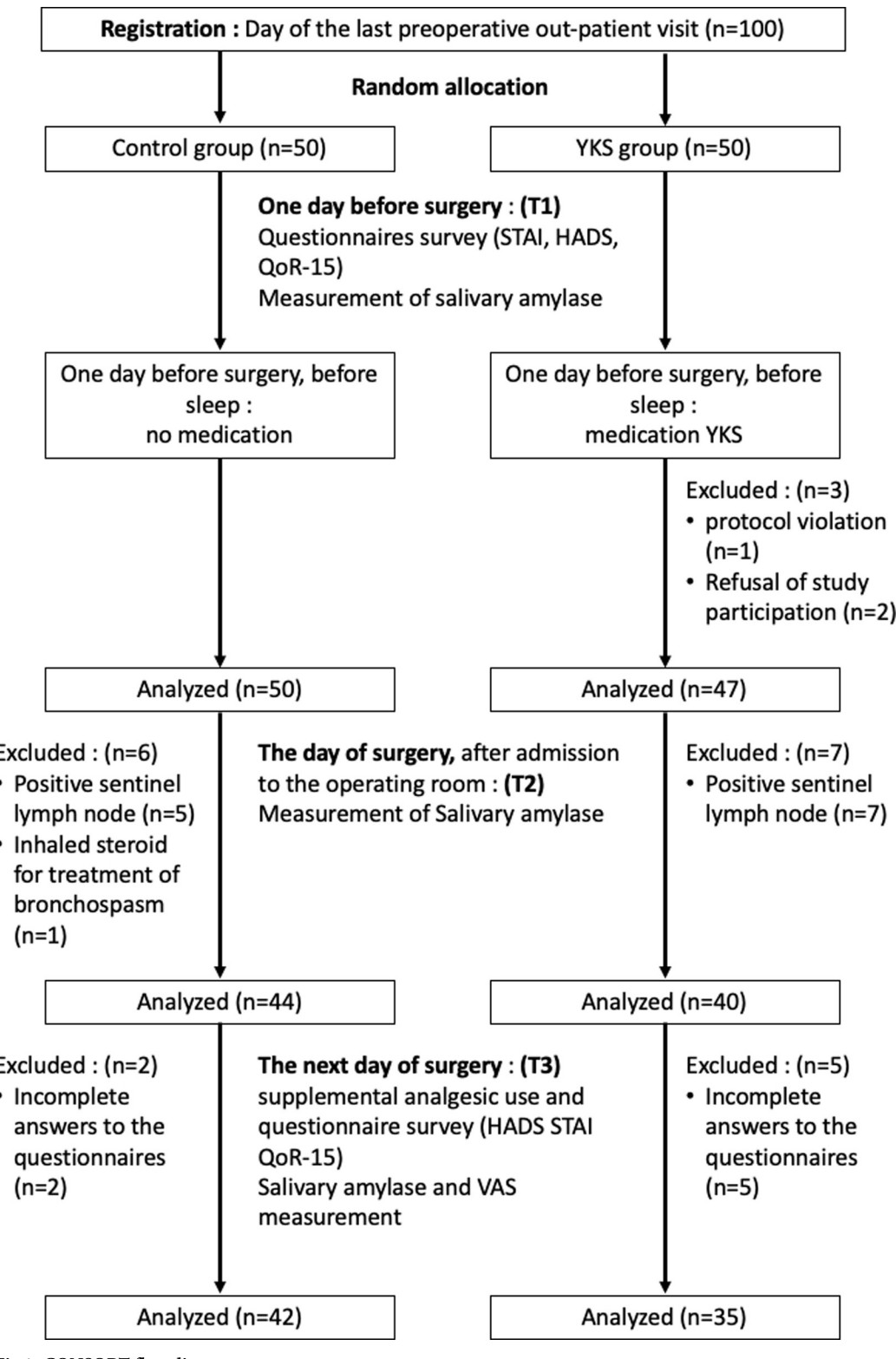

**Fig 1. CONSORT flow diagram.**

We assessed HADS and STAI scores as subjective anxiety indicators pre-and postoperatively. HADS scale is a self-administered screening test developed by Zigmond et al. [20] for

patients with physical illnesses. It is considered as a purely psychological assessment that excludes the influence of physical symptoms. This questionnaire consists of 14 items; 7 were related to depression (HADS-D), and 7 were related to anxiety (HADS-A), with values ranging from 0 to 21, and higher scores indicating more severe anxiety and depression. STAI is a self-reported measure of relatively stable individual differences in anxiety [21]. The test consists of two subscales: the S-scale, which evaluates anxiety as a transitory state, and the T-scale, which evaluates anxiety as a personality trait, with each subscale comprising 20 items. The S-scale evaluates how the respondent feels at a particular time due to stress owing to a perceived threat, while the T-scale evaluates how the respondent anticipates feeling during a hypothetical stressful situation. STAI is assessed on a scale of 20 to 80 points for both the T- and S-scales.

QoR questionnaires measure the recovery quality on multiple scales through patient-reported simple assessments of perioperative outcomes, including well-being, emotion, physical function, patient assistance, and pain [22]. In this study, we chose the QoR-15, which was previously utilized for breast cancer surgery [23]. QoR-15 was calculated by asking answers scored on a scale of 0 to 10, with the total score ranging from 0 to 150.

The 100-mm VAS score was used to measure subjective pain intensity and was assessed only once postoperatively at T3 (Fig 1). The VAS in this study had a 10-cm-long black line that was shown to the patient, with the far-left corresponding to "no pain" and the far-right corresponding to "the greatest pain imaginable"; this scale was used to indicate how much pain the patients were currently experiencing.

General anesthesia was induced by propofol target-controlled infusion (TCI) (target blood concentration 4.0 $\mu g \times ml^{-1}$), remifentanil continuous infusion (0.5 $\mu g \times kg^{-1}$ $min^{-1}$), and a 0.6-$mg \times kg^{-1}$ bolus injection of rocuronium to facilitate the insertion of the ProSeal Laryngeal Mask Airway (Teleflex Medical Japan, Tokyo, Japan). Anesthesia was maintained by propofol TCI, and the rate of remifentanil infusion was adjusted in the range of 0.05–0.5 $\mu g \times kg^{-1} \times min^{-1}$. Heart rate and blood pressure were maintained within 20% of their baseline values.

All breast tissues were removed during total mastectomy, whereas some breast tissues with tumors were removed during partial mastectomy. The surgeon also removed one sentinel lymph node in the latter operation. Lymph nodes were subsequently tested by intraoperative rapid pathological examination to evaluate whether they contained any cancer cells. If sentinel node metastasis was negative, the wound was closed, and the study continued; if metastasis was positive, the patient was excluded as axillary lymph node clearance was required.

For postoperative analgesia, 0.001% of epinephrine containing 1% lidocaine was injected into the surgical site directly before incision, and 1000 mg of acetaminophen was administered intravenously at the time of wound closure. If the patient complained of pain after awakening, 100 mg of tramadol was administered intravenously, and if the pain was still severe, 50 mg of diclofenac was additionally administered.

We used perioperative sAA to objectively measure anxiety and the HADS and STAI scores as subjective measures of anxiety. Changes in sAA were considered primary outcomes, and those in HADS, STAI, QoR-15, and VAS were considered secondary outcomes.

We determined our sample size based on the following criteria using G* power version 3.1.7 [24]. The mean (standard deviation) sAA score in a previous study was 37 (12). A difference in the sAA score of 15 points was considered statistically significant, and the sAA score for pain on postoperative day 1 was 25. Thus, a power analysis using a type I error estimate of 5% and a power of 80% indicated that a sample size of 40 patients per group was needed to detect this difference. Therefore, we decided to enroll 50 patients in each arm, considering possible dropouts.

For continuous variables, data were expressed as mean (standard deviation) after evaluating normality. Differences in values between pre- and postoperative assessments for each participant were expressed as mean [95% confidence interval] values. sAA scores were log-

transformed before the test to obtain a normal distribution. A Bartlett test was performed to confirm the equivalence of variances before analysis.

sAA measures data at three-time points. Therefore, the analysis of variance (ANOVA) performed on sAA measurements had two factors, that is, intervention (YKS and CNT) and time (pre-, directly post-surgery, and next day). The remaining data were measured at only two time points: pre-surgery and next day, as measurements could not be performed immediately after postoperatively.

Therefore, we described the ANOVA for sAA and other measures separately. Additionally, a repeated measures ANOVA was used to assess differences between HADS and STAI scores as intervention×time interaction effects. A multiple comparison analysis for intervention × time interaction effects were performed using Holm's modified sequentially rejective Bonferroni procedure. Significant and insignificant results were then followed up with a generalized eta squared estimation of effect size specifically designed for a repeated measures ANOVA. Confidence intervals for effect sizes were calculated using the bootstrap method. We used Welch's t-test to compare demographic data and VAS scores and Fisher's exact test to compare categorical data. All tests were two-tailed, with an alpha level of 0.05 indicating statistical significance. All analyses were conducted using R version 4.0.0 [25] and R function "anovakun" version 4.8.5. Data were visualized using violin plots [26], as implemented in Graphpad Prism 8 for macOS (Graphpad software, San Diego, CA). QoR-15 was not used in this analysis because a serious error was discovered at the time of analysis after measurement was completed, making the values of scale scores and total scores unreliable. Moreover, as dropout occurred after data acquisition, the sample size did not reach the calculated sample size. Hence, a post-hoc power analysis was conducted using G* power version 3.1.7.

## Results

### Recruitment and baseline characteristics

The participant flowchart is shown in Fig 1. The remaining 35 patients in the YKS group and 42 patients in the CNT groups were analyzed. Demographic characteristics are presented in Table 1. There were no significant differences in age, BMI, ASA, implementing facilities, and surgical procedure between both groups.

### Effects of YKS on perioperative anxiety and postoperative pain

sAA scores were significantly lower in the YKS group than in the CNT group at T2 (YKS, 0.88 (0.42); CNT, $1.14 \pm 0.49$; F [2,150] = 3.76; p = 0.03; and $g\eta^2$ = 0.017). Moreover, the sAA score

**Table 1. Patient demographic details.**

|  | YKS (n = 35) | CNT (n = 42) | P-value |
|---|---|---|---|
| Age, year (SD) | 49 (6.2) | 48 (5.7) | 0.72 |
| BMI (SD) | 20.9 (3.3) | 21.7 (2.8) | 0.09 |
| ASA (1:2) | 21:22 | 21:23 | 0.25 |
| Facility (university: cancer center) | 22:21 | 21:27 | 0.88 |
| Procedure (partial resection: total mastectomy) | 22:21 | 21:27 | 0.88 |

YKS, Yokukansan; CNT, control; SD, standard deviation; BMI, body mass index; and ASA, American Society of Anesthesiologists

Data are presented as mean (standard deviation) in each group and were analyzed using a t-test (age, BMI), and Wilcoxon rank sum test (ASA, implementing facilities, and surgical procedure).

at T2 was significantly lower than those at T1 and T3 in the YKS group (T1, 1.14 (0.55); T2, 0.88 (0.42); T3, 1.07 (0.54); T1 vs. T2: p = 0.03; T2 vs. T3: p = 0.1; and T1 vs. T3: p = 1.0), yet there were no significant differences in the sAA scores of the CNT group between all time-points (T1, 1.08 (0.50); T2, 0.14 (0.49); T3, 1.20 (0.49); T1 vs. T2: p = 1.0; T2 vs. T3: p = 1.0; and T1 vs. T3, 0.43; Fig 2). Results of the post-hoc power analysis showed slight high power (0.708) but did not reach the recommended power of 0.8.

Regarding the HADS-A score, there was an interaction effect between groups and time, with the YKS group showing a tendency towards higher scores than the CNT group; however, this difference was not statistically significant (F [1,75] = 3.08; p = 0.083; gη² = 0.01, as assessed by a two-way ANOVA). HADS-A scores were reduced postoperatively in both groups. However, the corresponding decline was greater in the YKS group than in the CNT group (YKS: T1, 7 [6–10]; T3: 6 [3–7]; T3-T1 difference, -2.77 [-1.48–-4.06]; p <0.001; and CNT: T1, 6 [5–9.25]; T3, 5 [3–8.25]; T3-T1 difference, -1.43 [-0.25–-2.61]; p = 0.011, as assessed by Holm's multiple comparison test, Fig 3).

Regarding the STAI-T score, there was also an interaction between groups and time, with the YKS group showing higher scores than the CNT group at T1 and T3 (F [1,75] = 7.26; p = 0.009; gη² = 0.011, as assessed by a 2-way ANOVA). The STAI-T score significantly decreased postoperatively in the YKS group, but not in the CNT group (YKS: T1, 47 [41–53]; T3, 44 [36–48]; T3-T1 difference, -4.23 [-1.51–-6.95]; p <0.001; CNT: T1, 45.5 [36.75–52.25]; T3, 44 [36.75–54.5]; T3-T1 difference, 0.12 [-2.36–2.60], p = 0.92, as assessed by Holm's multiple comparison test, Fig 4).

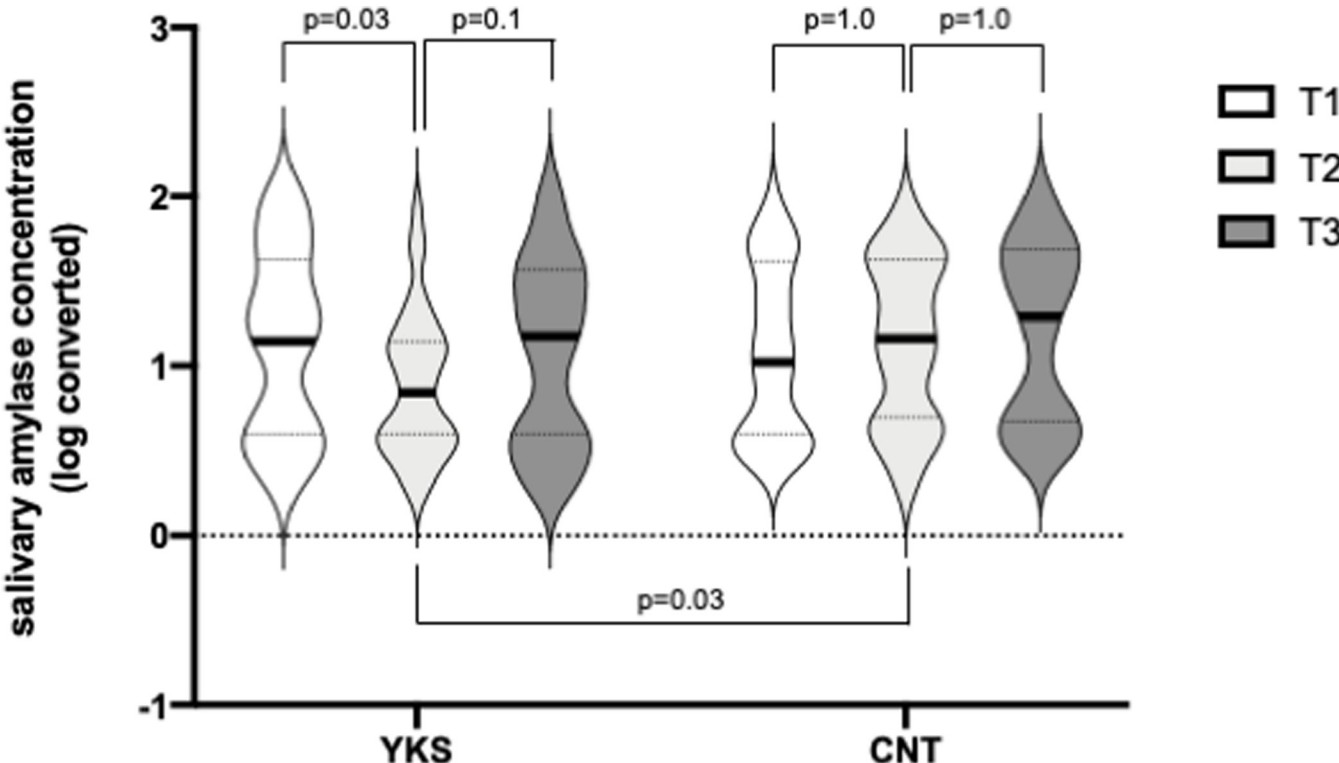

**Fig 2. Salivary-α-amylase activity.** Comparison of salivary-α-amylase activity in the Yokukansan (YKS) (a) and control (CNT) (b) groups by violin plots. Plot width is scaled to data distribution. Horizontal bars represent medians, and dashed lines represent 25th and 75th percentiles. The vertical axis represents logarithmically converted values. Data were analyzed by a two-way repeated-measures ANOVA and pre-specified follow-up tests of primary outcomes. Using a modified sequentially rejective Bonferroni test, a primary outcome with a p-value <0.05 was considered statistically significant.

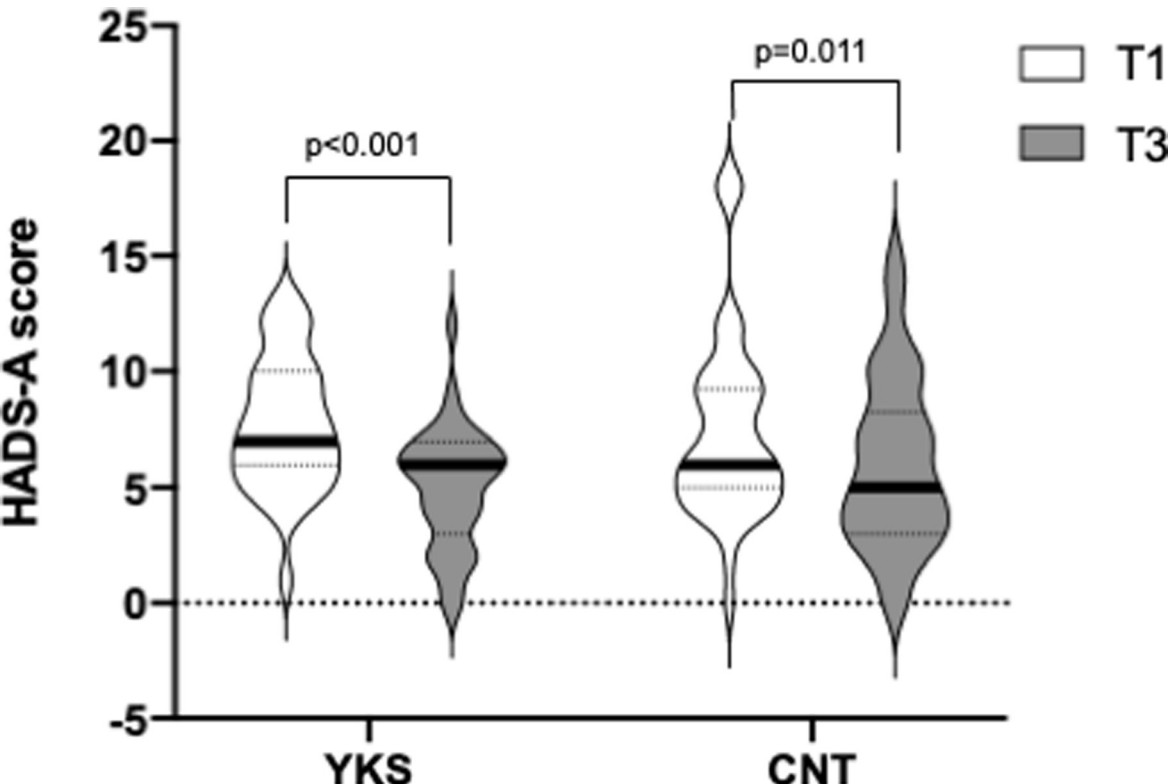

**Fig 3. Hospital Anxiety and Depression Scale-Anxiety (HADS-A) score.** Comparison of anxiety levels at T1 and T3, as assessed by the HADS-A score in the Yokukansan (YKS) and control (CNT) groups by violin plots. The vertical axis represents the score, and the plot width is scaled to data distribution. Horizontal bars represent medians, and dashed lines represent 25th and 75th percentiles. Data were analyzed by a two-way repeated-measures ANOVA. Using a sequentially rejective Bonferroni test, a primary outcome with a p-value <0.05 was considered statistically significant.

Contrarily, there were no significant differences in HADS-D, STAI-S, and VAS scores between both groups (Table 2). The possible side effects of a single dose of YKS include appetite loss, nausea, diarrhea, and rash. We investigated the presence of such side effects when the corresponding anesthesiologist visited the patient's room to measure postoperative amylase and collect questionnaires. There were no adverse events.

## Discussion

This study evaluated the effects of YKS on perioperative anxiety and postoperative pain using sAA, HADS, STAI, and VAS scores. Preoperative YKS administration significantly reduced thepreoperative sAA scores, but not one day postoperative scores. Furthermore, YKS affected HADS and STAI scores.

In a previous study, STAI and HADS were compared as measures of perioperative anxiety and were shown to correspond with anxiety on VAS [27]. In this study, two of these three indicators were used, and in both groups, postoperative HADS-A scores were lower than preoperative scores, possibly reflecting reassurance among patients upon the completion of surgery. However, the decrease in the HADS-A score was larger in the YKS group than in the CNT group. Postoperative STAI scores were also reduced in both groups; however, postoperative STAI-T scores significantly decreased only in the YKS group.

To the best of our knowledge, this is the first superiority trial to investigate the effects of YKS on anxiety using both subjective and objective rating scales. Our results suggested that

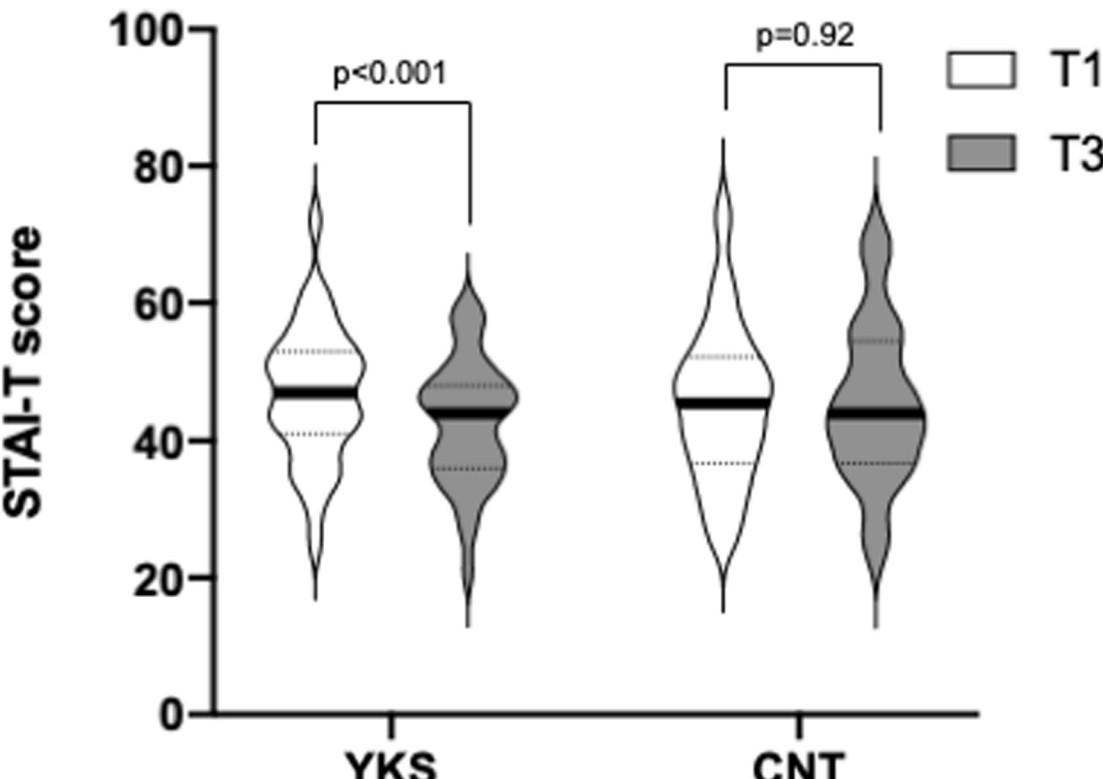

**Fig 4. State-Trait Anxiety Inventory-Trait (STAI-T) score.** Comparison of anxiety levels at T1 and T3, as assessed by the STAI-T score in the Yokukansan (YKS) and control (CNT) groups by violin plots. The vertical axis represents the score, and the plot width is scaled to data distribution. Horizontal bars represent medians, and dashed lines represent 25th and 75th percentiles. Data were analyzed by a two-way repeated measures ANOVA. Using a sequentially rejective Bonferroni test, a primary outcome with a p-value <0.05 was considered statistically significant.

two doses of YKS produced immediate preoperative physiological and psychological changes and confirmed that YKS effectively reduces perioperative anxiety.

YKS is used as an anxiolytic for irritability, restlessness, and insomnia and may have the pharmacological properties of antidepressant drugs. This effect may be explained by basic research showing that YKS affects the serotonin system in nervous tissue [11, 28]. Further, a previous large-scale study has shown that antidepressants improved trait anxiety [29]; our

**Table 2. Results of all questionnaires.**

| | YKS (n = 35) | | CNT (n = 42) | | P-value | | |
|---|---|---|---|---|---|---|---|
| | T1 | T3 | T1 | T3 | group | time | interaction |
| HADS-Anxiety | 7.74 (2.84) | 4.97 (2.58) | 7.48 (4.00) | 6.05 (3.69) | 0.55 | <0.01 | 0.08 |
| HADS-Depression | 4.71 (2.82) | 5.17 (3.16) | 6.10 (4.50) | 6.64 (4.45) | 0.09 | 0.11 | 0.88 |
| STAI-state | 47.26 (8.48) | 39.51 (8.33) | 50.33 (11.06) | 41.93 (10.71) | 0.17 | <0.01 | 0.75 |
| STAI-trait | 46.83 (9.75) | 42.60 (9.01) | 45.36 (11.36) | 45.48 (11.71) | 0.76 | 0.01 | 0.01 |
| VAS (pain) | | 1.4 (0.4971) | | 1.36 (0.49) | 0.70 | | |

YKS, Yokukansan; CNT, control; HADS, Hospital Anxiety and Depression Scale; STAI, State-Trait Anxiety Inventory; and VAS, visual analog scale

Data are presented as the mean (standard deviation) in each group. Data were analyzed using a t-test for VAS and a two-way repeated measures ANOVA for the remaining parameters.

result indicative of an improvement in STAI-T scores after YKS administration was consistent with the result of the aforementioned study.

Perioperative anxiety may prolong postoperative pain [2], which is one of the most unpleasant postoperative symptoms. CPSP, defined by the International Association for the Study of Pain as pain lasting >3 months postoperatively [30], is known to increase in patients with severe postsurgical pain. Moreover, depression, anxiety, satisfaction with postoperative pain management, and mental and physical stress are psychological risk factors for CPSP development and severity [31–33]. Currently, perioperative pain management is considered important for both patient satisfaction and CPSP prevention. Therefore, controlling perioperative anxiety and postoperative pain is crucial for CPSP prevention.

St. John's wort, valerian, kava, and passionflower are widely used for anxiety as alternatives to benzodiazepine; however, they have been associated with adverse effects, including increased bleeding and drug interactions [34]. There are no reports of drug interactions with YKS, and a previous study has suggested that YKS improves behavioral and psychological symptoms of dementia without serious side effects [35]. Hence, we hypothesized that YKS might be more beneficial than the aforementioned drugs. It is generally believed that the number of herbs included in a traditional Japanese prescription is directly proportional to the duration it takes for it to be effective. YKS contains seven herbs and is usually administered for a long duration. However, our results showed that YKS is effective even when taken for a short period, reducing the risk of side effects and patient's medication intake.

Basic research suggests that the pharmacological actions of YKS affect serotonin (5-HT) and the functions of the glutamate-mediated nervous system [11, 28, 36]. Serotonin is a neurotransmitter involved in the descending pain inhibitory system, weakening pain perception [37]. Moreover, excessive glutamate in the excitatory synaptic cleft is believed to be responsible for pain hypersensitivity [38]. Furthermore, the activation of the NMDA glutamate receptor is believed to be involved in hyperalgesia. YKS conventionally acts as an antagonist of NMDA receptors by binding to them and thus potentially treating neuropathic pain [39].

Although several clinical reports have suggested an analgesic effect of YKS, no relevant RCTs have been performed to confirm the same result. To the best of our knowledge, this is the first prospective RCT to investigate such effects. Contrary to our expectations, the effects of YKS on postoperative pain could not be demonstrated in this study. A possible reason for the lack of significant differences might have been the floor effect, with relatively low values in all patients. It may be necessary to investigate more invasive types of surgery to detect significant antinociceptive effects of YKS perioperatively. Moreover, the possibility that CPSP may be suppressed by the anxiolytic effect of YKS when taken postoperatively for a longer time remains to be elucidated.

This study has several limitations. First, it was not double-blinded. We could not form a placebo group because a placebo pill with the flavor and bitterness of Japanese herbal medicine is not available and would be difficult to produce. Conventionally, Japanese herbal medicines have a distinctive bitter taste and flavor. Sucrose and lactose, which are usually used as placebos, are different from Japanese herbal medicines, and people are familiar with the taste and thus cannot be used as placebos. Therefore, we deemed double blindness impossible, meaning that the patients were aware of their group. Thus, study results may not reflect the direct effects of YKS as the mere reassurance provided by its administration may have reduced anxiety and pain. Second, sAA may reflect stress-related physiological changes, though not anxiety specifically. However, as the psychological state of perioperative patients is often treated as "anxiety," we considered anxiety as a stress response based on Lazarus' psychological stress model [40]. Third, the number of subjects for analysis was less than 40 in the YKS group. In this study, more than 40 participants were initially assigned to each group, but after excluding dropouts,

42 and 35 participants in the YKS and CNT groups, respectively, were finally included in the analysis. The post-hoc power analysis yielded a high power of 0.708 but failed to reach the recommended power of 0.8. However, we attempted to provide reliable evidence from a small sample size by simultaneously reporting effect sizes and significant differences. Hence, this study was limited to 77 patients, and further large-scale investigations using placebos are needed to determine the effects of YKS on anxiety and pain.

## Conclusions

This RCT compared perioperative anxiety and postoperative pain in YKS-treated and control groups of women undergoing breast surgery. Our results suggested that YKS may reduce perioperative anxiety in patients undergoing surface surgery. Currently, there is no standard treatment for suppressing perioperative anxiety. While larger, double-blind, placebo-controlled studies are necessary, this study justifies further research into the potential of YKS for reducing perioperative anxiety.

## Supporting information

**S1 Checklist. CONSORT 2010 checklist of information to include when reporting a randomised trial**\*.
(DOC)

**S1 File. Research protocol (English version).**
(DOCX)

**S2 File. Research protocol (Japanese version).**
(DOCX)

**S3 File.**
(PDF)

**S4 File.**
(PDF)

**S1 Data.**
(XLSX)

**S2 Data.**
(XLSX)

## Acknowledgments

We would like to thank Editage (www.editage.jp) for English language editing.

## Author Contributions

**Conceptualization:** Moegi Tanaka, Tsunehiko Tanaka, Yoshinori Kamiya.

**Data curation:** Moegi Tanaka, Tsunehiko Tanaka, Misako Takamatsu, Chieko Shibue, Yuriko Imao, Takako Ando.

**Formal analysis:** Moegi Tanaka, Tsunehiko Tanaka, Yoshinori Kamiya.

**Funding acquisition:** Moegi Tanaka.

**Investigation:** Moegi Tanaka, Misako Takamatsu, Chieko Shibue, Yuriko Imao, Takako Ando.

**Methodology:** Tsunehiko Tanaka, Yoshinori Kamiya.

**Project administration:** Moegi Tanaka.

**Supervision:** Hiroshi Baba.

**Validation:** Tsunehiko Tanaka, Misako Takamatsu, Hiroshi Baba, Yoshinori Kamiya.

**Writing – original draft:** Moegi Tanaka, Tsunehiko Tanaka, Yoshinori Kamiya.

**Writing – review & editing:** Misako Takamatsu, Chieko Shibue, Yuriko Imao, Takako Ando, Hiroshi Baba.

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
