## [Decision Letter · Decision Letter 0]

8 Jun 2021

PONE-D-21-08399

Effects of the Kampo medicine Yokukansan for perioperative anxiety and postoperative pain in women undergoing breast surgery: A randomized, controlled trial

PLOS ONE

Dear Dr. Kamiya,

Thank you for submitting your manuscript to PLOS ONE. After careful consideration, we feel that it has merit but does not fully meet PLOS ONE’s publication criteria as it currently stands. Therefore, we invite you to submit a revised version of the manuscript that addresses **all of the points** raised during the review process.

Please ensure that your revised version matches PLOS ONE’s publication criteria.

We look forward to receiving your revised manuscript.

Kind regards,

Johannes Fleckenstein

Academic Editor

PLOS ONE

Journal Requirements:

2. "Please provide the specification sheet and lot number of the Tsumura Yokukansan Extract Granules (TJ-54, Tsumura & Co., Tokyo, Japan) used in this study, showing the full composition analysis of the product.

3.We note that you have indicated that data from this study are available upon request. PLOS only allows data to be available upon request if there are legal or ethical restrictions on sharing data publicly. For information on unacceptable data access restrictions, please see http://journals.plos.org/plosone/s/data-availability#loc-unacceptable-data-access-restrictions.

Reviewers' comments:

Reviewer's Responses to Questions

**Comments to the Author**

1. Is the manuscript technically sound, and do the data support the conclusions?

Reviewer #1: Partly

Reviewer #2: Yes

Reviewer #3: No

2. Has the statistical analysis been performed appropriately and rigorously? 

Reviewer #1: Yes

Reviewer #2: Yes

Reviewer #3: No

3. Have the authors made all data underlying the findings in their manuscript fully available?

Reviewer #1: Yes

Reviewer #2: Yes

Reviewer #3: Yes

4. Is the manuscript presented in an intelligible fashion and written in standard English?

Reviewer #1: No

Reviewer #2: Yes

Reviewer #3: Yes

5. Review Comments to the Author

Reviewer #1: YKS is used for Neurosis, insomnia, night cry in children, and peevishness in children in Japan. The authors have applied YKS for anxiety in the patients undergoing surgery. This randomized controlled study showed the effect of traditional Japanese medicine Yokukansan (YKS) for reducing perioperative anxiety in the patients undergoing surface surgery.

This study has novelty of YKS on the treatment in clinical setting. However, there are some lacks of study information. I comments some points as follows.

Introduction

The author described the side effects of benzodiazepines or antidepressants as follows in the Introduction section.

Benzodiazepines effectively reduce perioperative anxiety; however, they are associated with undesirable sedative effects [6]. Furthermore, antidepressants and other anxiolytics are not suitable for perioperative administration due to their ineffectiveness and significant side effects [7].

Please describe the detail of the dis advantage of these medicines.

Method

A detailed description of the intervention chemicals is needed. The amounts of crude drugs included in YKS extract should be explained in the Method section.

This study design seemed open label study. If not, the method of investigator blinding should be explained.

TJ-54 is indicated for the relief of the following symptoms of those patients with delicate constitution and nervousness: Neurosis, insomnia, night cry in children, and peevishness in children in Japanese National Health Insurance System. In the present study, YKS has prescribed for anxiety. Was YKS prescribed on the Japanese National Health Insurance System or administered with research funds? Please clarify how intervention was performed.

Result and interpretation

Patients can understand their intervention with or without YKS. Placebo effects is highly affected to the result. Please discuss more about these limitation.

The author showed the result of no side effects in the present study. However, there is no explanation about the evaluation of adverse event and side effect. Please add the explabation how the investigator evaluate adverse event and side effect.

Please add the detail data of QoR-15.

Reviewer #2: This is a well-written report of a clinical trial. From a statistical point of view, the investigators used minimization on two variables to randomize (?). The question mark indicates that it is not clear if they used a deterministic minimization (Taves) or a randomized design (Pocock-Simon). Please clarify. Also, if there are only 2 variables, why use minimization? Why not just stratify? Please explain. The sample size computation is appropriate, but they were not able to achieve the requisite sample size. In the discussion, I would like to see a post-hoc power analysis using the observed parameter estimates, rather than the guesses in the sample size section. Finally, since it was not double-blinded, how was the outcome assessed? Who administered the questionnaires? Was it someone who was unaware of the "treatment"?

Reviewer #3: The authors have shown the efficacy of yokukansan on anxiety using both subjective and objective rating scales. We think that it is very interesting study; however, the problems listed below will need to be addressed before the paper is suitable for publication.

Major points

Although the authors used changes in sAA, HADS, and STAI as primary outcomes, it would be best to use only one thing as the primary outcome. Further, the sample size was for sAA in this study. If you decide on a primary outcome other than sAA, the sample size will become quite different.

The authors planned for yokukansan to be taken two times before sleep on the night before surgery and 2h before the induction of anesthesia. Was the duration of the first two times the same in this study? Further, why did you decide to take yokukansan 2hr before the induction of anesthesia?

Minor points

1 In the analysis of HADS, is there a statistically significant difference between the two groups over time?

2 The figure legends should be written after the references, please check that the formatting follows the journal’s instructions for authors.

6. PLOS authors have the option to publish the peer review history of their article (what does this mean?). If published, this will include your full peer review and any attached files.

Reviewer #1: **Yes: **Shin Takayama

Reviewer #2: No

Reviewer #3: No

---

## [Author Response · Author response to Decision Letter 0]

7 Jul 2021

Reviewer #1: YKS is used for Neurosis, insomnia, night cry in children, and peevishness in children in Japan. The authors have applied YKS for anxiety in the patients undergoing surgery. This randomized controlled study showed the effect of traditional Japanese medicine Yokukansan (YKS) for reducing perioperative anxiety in the patients undergoing surface surgery.

This study has novelty of YKS on the treatment in clinical setting. However, there are some lacks study information. I comment some points as follows.

Introduction

The author described the side effects of benzodiazepines or antidepressants as follows in the Introduction section.

Benzodiazepines effectively reduce perioperative anxiety; however, they are associated with undesirable sedative effects [6]. Furthermore, antidepressants and other anxiolytics are not suitable for perioperative administration due to their ineffectiveness and significant side effects [7].

Please describe the detail of the disadvantage of these medicines.

We thank reviewer #1 for the valuable comments. We have added specific examples of the disadvantages of using antidepressants and other anxiolytics in the introduction as follows (P5, L2–5): “Benzodiazepines effectively reduce perioperative anxiety; however, they are associated with undesirable sedative effects [6]. Furthermore, antidepressants and other anxiolytics are not suitable for perioperative administration due to their ineffectiveness and side effects, including nausea, thirst, and drowsiness [7].”

Method

A detailed description of the intervention chemicals is needed. The amounts of crude drugs included in YKS extract should be explained in the Method section.

We have added extensive details on yokukansan as follows (P9, L7-10): “Patients in the YKS group received two doses of 2.5 g Tsumura Yokukansan Extract Granules (TJ-54, Tsumura & Co., Tokyo, Japan) before sleeping on the night before surgery and 2 h before inducing anesthesia, and 2.5 g of YKS contains 1.083 g of a dry extract of mixed herbal medicines in the following proportions: Atractylodes Lancea Rhizome 19.5%, Poria Sclerotium 19.5%, Cnidium Rhizome 14.6%, Uncaria Hook 14.6%, Japanese Angelica Root 14.6%, Bupleurum Root 9.7%, and Glycyrrhiza 7.3%.”

This study design seemed open label study. If not, the method of investigator blinding should be explained.

As reviewer #1 indicated, this clinical study is a prospective open label trial in which the assessor was single-blinded, as shown in head of the method section (P7, L9-10). 

TJ-54 is indicated for the relief of the following symptoms of those patients with delicate constitution and nervousness: Neurosis, insomnia, night cry in children, and peevishness in children in Japanese National Health Insurance System. In the present study, YKS has prescribed for anxiety. Was YKS prescribed on the Japanese National Health Insurance System or administered with research funds? Please clarify how intervention was performed.

In this study, YKS was prescribed on the Japanese National Health Insurance System.

Result and interpretation

Patients can understand their intervention with or without YKS. Placebo effects is highly affected to the result. Please discuss more about these limitations.

We completely agree with the notion of reviewer #1. Since this is an open-label clinical trial, the influence of the placebo effect cannot be excluded. We also were not able to exclude the possibility of the fact that the patients took the medication affected their amylase levels or questionnaire results. Based on the results of this study, a more rigorous, double-blind study should be conducted. 

Several other studies used sucrose or lactose powder as placebo medication, but since many Japanese people understand the unique flavor of Kampo medicine, even if these were used as placebo medicine, patients would immediately know that this was not Kampo medicine and would not be blinded. In other words, we thought it was less legitimate to use sucrose or lactose as placebo medication and decided to have the control group drink only water as an open-label study. We have added this opinion in the limitations section (P26 L9-P27L1).

The author showed the result of no side effects in the present study. However, there is no explanation about the evaluation of adverse event and side effect. Please add the explanation how the investigator evaluates adverse event and side effect.

The possible side effects of a single dose of yokukansan include appetite loss, nausea, diarrhea, and rash. We investigated the presence of such side effects during the visit of the anesthesiologist to the patient’s room for measuring postoperative amylase levels and collecting questionnaires. There were no adverse events (P21 L5-L8).

Please add the detail data of QoR-15.

Thank you for your advice. Based on your suggestions, we reviewed the data in detail to perform a reanalysis and found that there was a serious mistake in the QoR-15 questionnaire; therefore, we deemed analyzing the corresponding data to be inappropriate Since all outcomes were pre-registered, it was clearly stated that QoR-15 was excluded from the analysis. QoR-15 was described as a measured item in the methods section. We have added the corresponding statements in the methods section (P15 L14-P16L1).

Reviewer #2: This is a well-written report of a clinical trial. From a statistical point of view, the investigators used minimization on two variables to randomize (?). The question mark indicates that it is not clear if they used a deterministic minimization (Taves) or a randomized design (Pocock-Simon). Please clarify. 

We are glad to receive positive comments from reviewer #2. In this study, our allocation algorithm was based on the Pocock & Simon method, and we used three variables for dynamic allocation: age, HADS, and institution. We did not use stratification because we wanted to make grouping as simple as possible.

Also, if there are only 2 variables, why use minimization? Why not just stratify? Please explain. The sample size computation is appropriate, but they were not able to achieve the requisite sample size. In the discussion, I would like to see a post-hoc power analysis using the observed parameter estimates, rather than the guesses in the sample size section.

Moreover, following your suggestion, we tested the post-hoc power analysis using G* power version 3.1. The result of the power analysis was 0.708, which was deemed to be somewhat a high-power result; however, the analysis did not reach the recommended power of 0.8. As described in the main text, although we reached the predetermined number of cases, there were more dropouts than expected. We have added the results of the post-hoc analysis to the text and added a description in the limitation section about the fact that the power of the test was less than 0.8 (P18 L4-5).

Finally, since it was not double-blinded, how was the outcome assessed? Who administered the questionnaires? Was it someone who was unaware of the "treatment"?

The questionnaire was administered by an anesthesiologist who was not involved in the allocation of the study groups and not informed to which group the patients were assigned (P9 L15-P10L2).

Reviewer #3: The authors have shown the efficacy of yokukansan on anxiety using both subjective and objective rating scales. We think that it is very interesting study; however, the problems listed below will need to be addressed before the paper is suitable for publication.

We thank reviewer #3 for the positive comments. We will answer the issues raised by reviewer #3 as much as possible as follows:

Major points

Although the authors used changes in sAA, HADS, and STAI as primary outcomes, it would be best to use only one thing as the primary outcome. Further, the sample size was for sAA in this study. If you decide on a primary outcome other than sAA, the sample size will become quite different.

Thank you for your advice. As you suggested, we set sAA as the primary outcome and questionnaire results as secondary outcomes. We calculated the sample size using differences in sAA as already shown in the sample size calculation (P13 L12-15). 

The authors planned for yokukansan to be taken two times before sleep on the night before surgery and 2h before the induction of anesthesia. Was the duration of the first two times the same in this study? Further, why did you decide to take yokukansan 2hr before the induction of anesthesia?

In a previous study (Evid Based Complement Alternat Med. 2014;2014: 965045, ref. #12), YKS was administered once 5 h before surgery. However, in cases in which surgery is conducted in the morning, YKS is orally administered very early in the morning, two hours before prohibiting water. However, since the previous study used drugs that were non-inferior to benzodiazepines, we aimed to administer a slightly higher dose of YKS to clarify its effect in this superiority study. Nevertheless, since prolonging the duration of medication would increase the number of dropouts, we started medication administration after the corresponding patient was admitted to the hospital to ensure that the medication was administered twice.

Minor points

1 In the analysis of HADS, is there a statistically significant difference between the two groups over time?

We found that there were no significant differences in the main effect, but there were significant differences in the interaction.

2 The figure legends should be written after the references, please check that the formatting follows the journal’s instructions for authors.

The journal guidelines verify that figure legends should be included immediately after the citation of the corresponding figure in the text. Please see the corresponding guidelines below:

• Figure captions are inserted immediately after the first paragraph in which the figure is cited. Figure files are uploaded separately.

---

## [Decision Letter · Decision Letter 1]

22 Oct 2021

PONE-D-21-08399R1

Effects of the Kampo medicine Yokukansan for perioperative anxiety and postoperative pain in women undergoing breast surgery: A randomized, controlled trial

PLOS ONE

Dear Dr. Kamiya,

Thank you for submitting your manuscript to PLOS ONE. After careful consideration, we feel that it has merit but does not fully meet PLOS ONE’s publication criteria as it currently stands. Therefore, we invite you to submit a revised version of the manuscript that addresses the points raised during the review process.

We look forward to receiving your revised manuscript.

Kind regards,

Johannes Fleckenstein

Academic Editor

PLOS ONE

Journal Requirements:

Reviewers' comments:

Reviewer's Responses to Questions

**Comments to the Author**

1. If the authors have adequately addressed your comments raised in a previous round of review and you feel that this manuscript is now acceptable for publication, you may indicate that here to bypass the “Comments to the Author” section, enter your conflict of interest statement in the “Confidential to Editor” section, and submit your "Accept" recommendation.

Reviewer #1: All comments have been addressed

Reviewer #2: All comments have been addressed

Reviewer #3: (No Response)

2. Is the manuscript technically sound, and do the data support the conclusions?

Reviewer #1: Yes

Reviewer #2: (No Response)

Reviewer #3: Yes

3. Has the statistical analysis been performed appropriately and rigorously? 

Reviewer #1: Yes

Reviewer #2: (No Response)

Reviewer #3: Yes

4. Have the authors made all data underlying the findings in their manuscript fully available?

Reviewer #1: No

Reviewer #2: (No Response)

Reviewer #3: Yes

5. Is the manuscript presented in an intelligible fashion and written in standard English?

Reviewer #1: Yes

Reviewer #2: (No Response)

Reviewer #3: Yes

6. Review Comments to the Author

Reviewer #1: In the discussion with Reviewers, the authors revised as follows.

Changes in sAA were considered as primary outcomes, and those in HADS, STAI, QoR-15, and VAS were considered as secondary outcomes.

However, in the UMIN registration, the authors declare that Primary outcomes as “subjective evaluation items: HADS( Hospital anxiety and depression scale)” and “objective evaluation items: salivaly amylase activity” and Secondary outcomes as “STAI(State Trait Anxiety Inventory)”, QoR (quality of recovery) score”, and “VAS score of pain”.

This study is designed clinical trial, if you change this point, amendment of trial design is needed.

Reviewer #2: (No Response)

Reviewer #3: Major points

Although the authors used changes in sAA, HADS, and STAI as primary outcomes, it

would be best to use only one thing as the primary outcome. Further, the sample size

was for sAA in this study. If you decide on a primary outcome other than sAA, the

sample size will become quite different.

OK

Thank you for your advice. As you suggested, we set sAA as the primary outcome and

questionnaire results as secondary outcomes. We calculated the sample size using

differences in sAA as already shown in the sample size calculation (P13 L12-15).

The authors planned for yokukansan to be taken two times before sleep on the night

before surgery and 2h before the induction of anesthesia. Was the duration of the first

two times the same in this study? Further, why did you decide to take yokukansan 2hr

before the induction of anesthesia?

In a previous study (Evid Based Complement Alternat Med. 2014;2014: 965045, ref.

#12), YKS was administered once 5 h before surgery. However, in cases in which

surgery is conducted in the morning, YKS is orally administered very early in the

Powered by Editorial Manager® and ProduXion Manager® from Aries Systems Corporation

morning, two hours before prohibiting water. However, since the previous study used

drugs that were non-inferior to benzodiazepines, we aimed to administer a slightly

higher dose of YKS to clarify its effect in this superiority study. Nevertheless, since

prolonging the duration of medication would increase the number of dropouts, we

started medication administration after the corresponding patient was admitted to the

hospital to ensure that the medication was administered twice.

OK

Minor points

1 In the analysis of HADS, is there a statistically significant difference between the two

groups over time?

We found that there were no significant differences in the main effect, but there were

significant differences in the interaction.

OK

2 The figure legends should be written after the references, please check that the

formatting follows the journal’s instructions for authors.

The journal guidelines verify that figure legends should be included immediately after

the citation of the corresponding figure in the text. Please see the corresponding

guidelines below:

•Figure captions are inserted immediately after the first paragraph in which the figure is

cited. Figure files are uploaded separately.

OK

The author responded appropriately to reviewers comments.

7. PLOS authors have the option to publish the peer review history of their article (what does this mean?). If published, this will include your full peer review and any attached files.

Reviewer #1: **Yes: **Shin Takayama

Reviewer #2: No

Reviewer #3: No

---

## [Author Response · Author response to Decision Letter 1]

1 Nov 2021

Reviewer #1: In the discussion with Reviewers, the authors revised as follows.

Changes in sAA were considered as primary outcomes, and those in HADS, STAI, QoR-15, and VAS were considered as secondary outcomes.

However, in the UMIN registration, the authors declare that Primary outcomes as “subjective evaluation items: HADS　(Hospital anxiety and depression scale)” and “objective evaluation items: salivary amylase activity” and Secondary outcomes as “STAI (State Trait Anxiety Inventory)”, QoR (quality of recovery) score”, and “VAS score of pain”.

This study is designed clinical trial, if you change this point, amendment of trial design is needed.

Response

We thank reviewer #1 for pointing out the ambiguity in the manuscript. In this clinical study, we needed re-registration with the Japan Registry of Clinical Trials (jRCT). In the modified registry, we set the value of sAA (objective outcome of perioperative anxiety) as the primary outcome, and questionnaire-assessed anxiety and perioperative pain (HADS, STAI, QoR-15, and VAS) were set as secondary outcomes. The jRCT has been reviewed by the secretariat of the Central Clinical Research Review Committee of Niigata University, which is a third-party organization and has a stricter protocol. This research was conducted in accordance with the research protocol registered in jRCT, which is generally consistent with the research protocol registered in UMIN. 

We have added the description about registration in jRCT and described primary and secondary outcomes in the materials and methods section (P8L109 – P9L113).

Reviewer #2: (No Response)

Reviewer #3: Major points

・Although the authors used changes in sAA, HADS, and STAI as primary outcomes, it would be best to use only one thing as the primary outcome. Further, the sample size was for sAA in this study. If you decide on a primary outcome other than sAA, the sample size will become quite different.

Response: Thank you for your the valuable suggestion. Accordingly, we set sAA as the primary outcome and questionnaire results as a secondary outcome. We calculated the sample size using differences in sAA as already shown in the sample size calculation (P14 L191-195). 

OK

・The authors planned for yokukansan to be taken two times before sleep on the night before surgery and 2h before the induction of anesthesia. Was the duration of the first two times the same in this study? Further, why did you decide to take yokukansan 2hr before the induction of anesthesia?

Response: In a previous study (Evid Based Complement Alternat Med. 2014;2014: 965045, ref. #12), Yokukansan was administered once 5 hours before surgery. However, in cases where surgery is conducted in the morning, YKS was orally administered very early in the morning, two hours before prohibiting water. However, since the previous study used drugs that were non-inferior to benzodiazepines, we aimed to administer a slightly higher dose of Yokukansan to clarify its effect in this superiority study. Nevertheless, since prolonging the duration of medication would increase the number of dropouts, we started administration of yokukansan after hospital admission of patients to ensure that the medication was administered twice.

OK

Minor points

1 In the analysis of HADS, is there a statistically significant difference between the two groups over time?

Response: We found no significant differences in the main effect, but there were significant differences in the interaction.

OK

2 The figure legends should be written after the references, please check that the formatting follows the journal’s instructions for authors.

The journal guidelines verify that figure legends should be included immediately after the citation of the corresponding figure in the text. Please see the corresponding guidelines below:

Response: Figure captions have been inserted immediately after the first paragraph whereeach figure is cited as mentioned in the submission guidelines. Figure files are uploaded separately.

OK

The author responded appropriately to reviewers’ comments. 

Response: We thank reviewer #3 for the effort in assessing our rebuttal letter.

---

## [Editor Report · Decision Letter 2]

12 Nov 2021

Effects of the Kampo medicine Yokukansan for perioperative anxiety and postoperative pain in women undergoing breast surgery: A randomized, controlled trial

PONE-D-21-08399R2

Dear Dr. Kamiya,

We’re pleased to inform you that your manuscript has been judged scientifically suitable for publication and will be formally accepted for publication once it meets all outstanding technical requirements.

Kind regards,

Johannes Fleckenstein

Academic Editor

PLOS ONE
---

## [Editor Report · Acceptance letter]

16 Nov 2021

PONE-D-21-08399R2 

Effects of the Kampo medicine Yokukansan for perioperative anxiety and postoperative pain in women undergoing breast surgery: A randomized, controlled trial 

Dear Dr. Kamiya:

I'm pleased to inform you that your manuscript has been deemed suitable for publication in PLOS ONE. Congratulations! Your manuscript is now with our production department. 

Kind regards, 

on behalf of

Priv.-Doz. Dr. Johannes Fleckenstein 

Academic Editor

PLOS ONE